**communications** engineering

# A deep learning framework based on structured space model for detecting small objects in complex underwater environments

Yaoming Zhuang [1,7] ✉, Jiaming Liu[1,7], Haoyang Zhao[1,2], Longyu Ma[1,2], Zirui Fang[3], Li Li[4], Chengdong Wu[1], Wei Cui[5] & Zhanlin Liu[6]

Regular monitoring of marine life is essential for preserving the stability of marine ecosystems. However, underwater target detection presents several challenges, particularly in balancing accuracy with model efficiency and real-time performance. To address these issues, we propose an innovative approach that combines the Structured Space Model (SSM) with feature enhancement, specifically designed for small target detection in underwater environments. We developed a high-accuracy, lightweight detection model—UWNet. The results demonstrate that UWNet excels in detection accuracy, particularly in identifying difficult-to-detect organisms like starfish and scallops. Compared to other models, UWNet reduces the number of model parameters by 5% to 390%, substantially improving computational efficiency while maintaining top detection accuracy. Its lightweight design enhances the model's applicability for deployment on underwater robots.

The ocean, which covers approximately 71% of Earth's surface, represents the largest biome on the planet. Marine organisms play a vital role in maintaining the delicate balance of this ecosystem[1–3]. They regulate the global climate through carbon sequestration and support biodiversity through complex food webs. Benthic marine organisms, such as holothurians and scallops, provide food and medicinal value while enhancing marine sediment health and productivity. For example, holothurians can increase seabed sediment productivity by up to 50%, and scallops can filter up to 200 l of water per hour. Therefore, regular and effective marine biological monitoring is crucial for assessing ecosystem status and implementing timely conservation measures.

Given the challenging conditions of the marine environment, especially in deep underwater regions, manual detection and recording are often difficult. Underwater robots have become essential for these tasks, but their computational power is restricted by hardware limitations, making real-time detection a critical requirement. Therefore, an efficient and lightweight framework for underwater biological detection is needed[4].

Recent research in object detection using deep learning has primarily focused on two approaches: single-stage detection algorithms and two-stage detection algorithms. Single-stage object detection algorithms, commonly based on the YOLO series, deliver real-time results by directly generating detection outputs. Enhancements to these algorithms typically involve adding attention mechanisms, improving feature extraction, or designing better feature fusion branches to improve the detection of small targets[5–8]. In contrast, two-stage object detection algorithms, such as the R-CNN series, first use a region proposal network (RPN) to generate candidate regions and then refine these regions using convolutional neural networks. Improving two-stage detectors involves enhancing the accuracy of candidate region generation, for example, by utilizing the self-attention mechanisms of Transformers to improve small target detection[9–12]. While two-stage algorithms outperform single-stage algorithms in accuracy, their high computational complexity and slower inference speed render them unsuitable for real-time monitoring tasks. Conversely, single-stage object detection algorithms are constrained by the limitations of convolutional neural networks (CNNs), which rely on fixed-stride convolutional windows for feature extraction. This approach restricts the receptive field at each layer, reducing the model's ability to capture global and long-distance dependencies[13–17]. In response to this challenge, researchers have increasingly explored the use of

[1]Faculty of Robot Science and Engineering, Northeastern University, Shenyang, China. [2]College of Information Science and Engineering, Northeastern University, Shenyang, China. [3]School of Information and Artificial Intelligence, Anhui Agricultural University, Hefei, China. [4]JangHo School of Architecture, Northeastern University, Shenyang, China. [5]Institute for Infocomm Research, Astar, Singapore. [6]AstrumU, Bellevue, WA, USA. [7]These authors contributed equally: Yaoming Zhuang, Jiaming Liu. ✉e-mail: zhuangyaoming@mail.neu.edu.cn

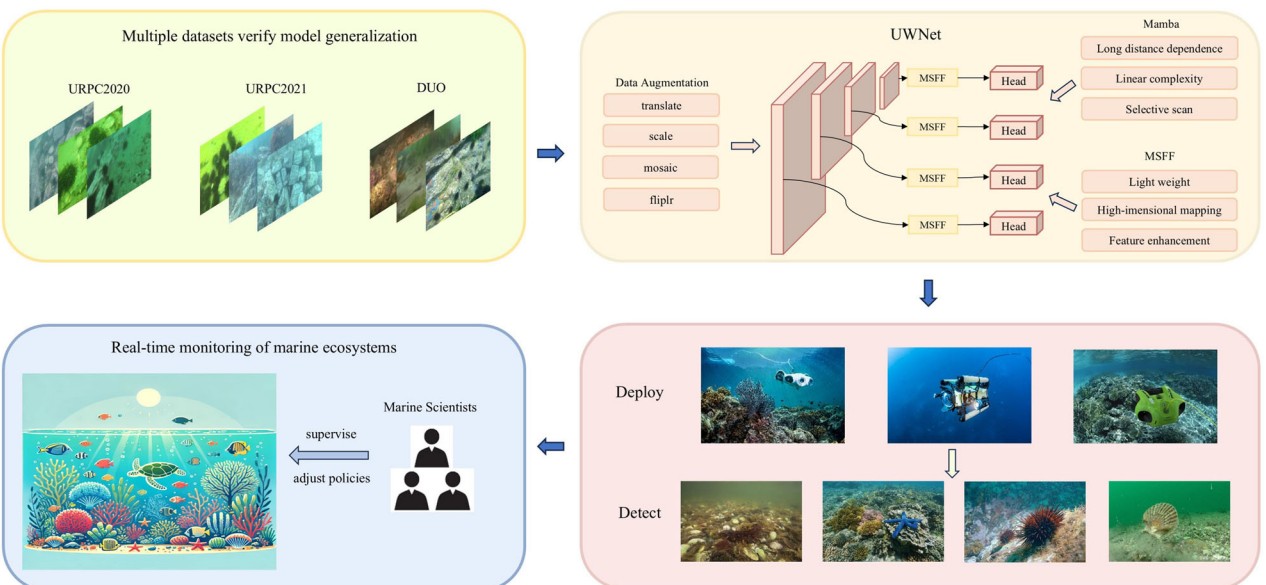

**Fig. 1 |** The pipeline of our work.

Transformer models in object detection. Transformer, with its self-attention mechanism, excels at capturing long-range dependencies. However, the computational complexity of self-attention scales quadratically with the input sequence length, resulting in substantial computational and memory demands. Moreover, Transformer models often struggle in few-shot learning scenarios, further limiting their applicability in certain domains[18–20].

To address the limitations of CNNs and Transformers, researchers have developed novel models such as the Mamba model[21]. Introduced in late 2023 and increasingly utilized for computer vision tasks in 2024, Mamba initially achieved success in image classification. It later expanded its applications to image segmentation and object detection, making meaningful contributions to advancements in computer vision technology[22–24]. The key advantages of Mamba are twofold: first, its global modeling capability, similar to that of Transformers, which effectively addresses the limitations of CNNs' receptive fields; second, its linear computational complexity compared to Transformers, making it highly suitable for resource-constrained scenarios.

Although Mamba has been applied to object detection tasks, our approach offers innovative insights into its application for underwater object detection. In this context, we introduce UWNet, a deep learning-based object detection network, which builds on the YOLOv8 framework and incorporates the Mamba model to enhance feature extraction. We apply the Mamba model to underwater target detection, aimed at improving the detection of small objects in complex underwater environments. Our multi-scale implicit feature fusion (MSFF) module, compared to current attention-based methods, offers a more lightweight solution for capturing small object features. By integrating Mamba into the network's backbone, we leverage its selective scanning mechanism to address the limitations of CNN's local window feature extraction and combine the advantages of Transformers with the linear computational efficiency of SSM. Experiments demonstrate that our method performs exceptionally well on underwater datasets, remarkably outperforming the latest object detection algorithms. It achieves state-of-the-art detection accuracy and accurately detects small underwater objects while maintaining a low parameter count. This makes it highly suitable for deployment on underwater robots. Our method shows substantial improvements over baseline models across four test sets (Test-A, Test-B, URPC2021, and DUO). Specifically, the mean Average Precision (mAP50) increased by 7.1%, 7.2%, 3.1%, and 4%, respectively, and the mAP50-95 increased by 4.8%, 5.1%, 3.9%, and 6%, respectively. Additionally, our model has a parameter count of only 6.67 million, which represents a reduction of 40%, 67%, 21%, 41%, 227%, and 390% compared to the latest mainstream object detectors, including YOLOv7, YOLOv8s, YOLOv10s, YOLOv11s, Mamba-YOLO-B, and RT-DETR.

UWNet effectively demonstrates the advantages of combining Mamba with YOLO for underwater small object detection. Its lightweight design makes it ideal for deployment on underwater robots, enabling efficient data collection and target detection. This enhances marine exploration and provides reliable support for ecological conservation. Figure 1 illustrates the pipeline of our study, presenting the key steps and processes involved throughout the research.

## Results

Underwater datasets often face challenges like poor image quality and monotonous background colors, making it harder to distinguish targets from their surroundings. Additionally, the prevalence of small targets and their frequent occlusion in underwater environments often hinder object detection networks from effectively capturing the key features of underwater targets during feature extraction. In this section, we introduce two innovative modules before presenting our proposed underwater object detection network, UWNet. We then present UWNet's performance on various underwater datasets and compare it with the results of other object detectors.

### Multi-scale implicit feature enhancement module

Attention mechanisms, including CBAM, ECA, SE, CA, and EMA[25–29], are widely used to enhance the accuracy of target detection models, especially for small objects. These methods improve feature extraction by focusing on spatial dimensions, channel dimensions, or their combination. Despite their effectiveness across diverse datasets, these mechanisms face limitations in underwater environments, which are often dominated by small targets. Over-reliance on channel or spatial features can lead the model to over-emphasize high-resolution background information, resulting in the extraction of irrelevant contextual details (as shown in Supplementary Table 1, which compares various attention mechanisms with the MSFF module).

To overcome these challenges and improve small target detection while minimizing computational costs, we introduce a lightweight multi-scale implicit feature enhancement module (MSFF). By utilizing advanced multi-view feature aggregation and element-wise multiplication for implicit feature enhancement[30], MSFF achieves a balanced extraction of fine-grained details and semantic information from input images.

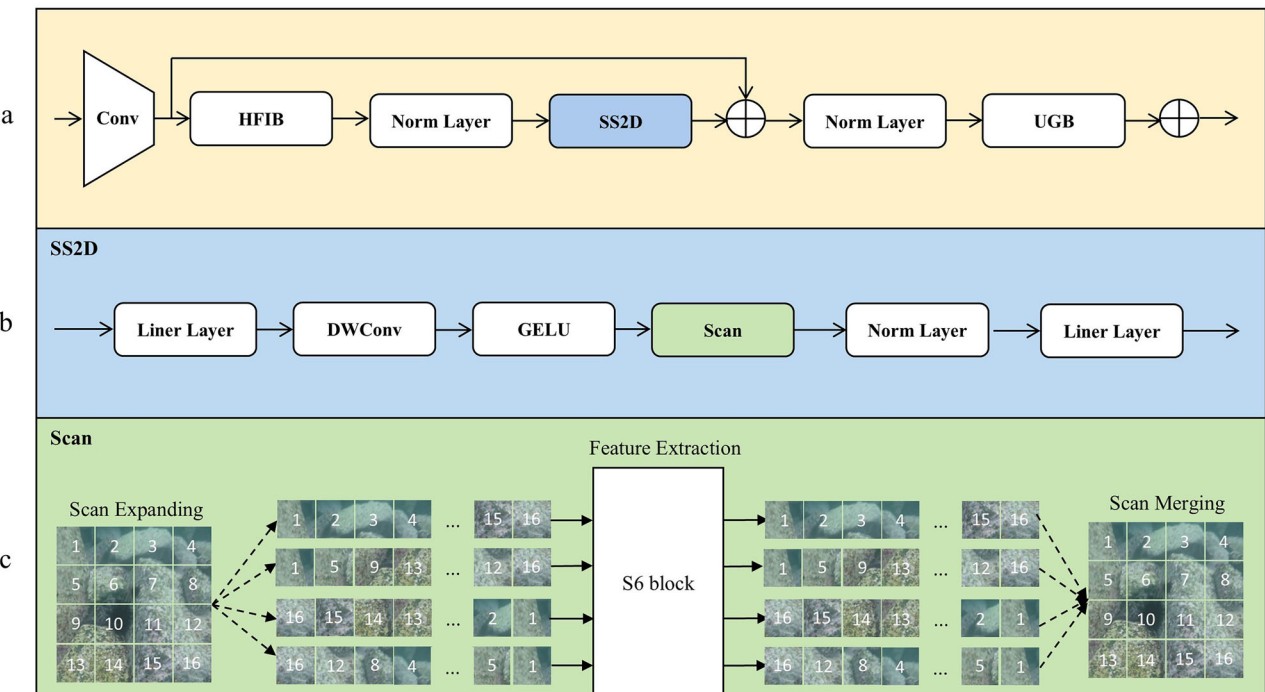

**Fig. 2 | Illustrates the design of the MSDBlock module and associated components in this paper. a** Schematic diagram of the MSDBlock module structure. **b** Flowchart of the SS2D module. **c** Core component of the SS2D module, the selective scanning feature.

Experimental results show that the MSFF module substantially improves model performance, especially in detecting small targets and handling complex scenes. Through its multi-scale implicit feature fusion mechanism, MSFF enables high-dimensional multi-scale integration of input feature maps, leading to a marked improvement in detection accuracy and robustness. This innovative module offers an effective solution for computer vision tasks, showcasing exceptional adaptability and performance in challenging scenarios.

### SSM-based feature extraction module

This study utilizes the Mamba model for underwater target detection, achieving notable improvements in both accuracy and efficiency. The Mamba model offers several key advantages: (1) it employs the HiPPO method to model long-distance dependencies, effectively overcoming the limitations of CNN's local receptive fields; (2) a selective scanning mechanism transforms state-space equation parameters into input parameters, enabling real-time adaptation to diverse input scenarios; (3) it achieves linear computational complexity, greatly reducing computational costs compared to the self-attention mechanism of Transformers.

To adapt the Mamba model for underwater target detection, we proposed MSDBlock, a feature extraction module based on CNN and SSM, as shown in Fig. 2a. Through experimental exploration, we found that relying solely on SSM is insufficient for capturing small target features in complex underwater environments. This limitation arises because SSM operates as a causal modeling method, akin to RNNs, where each output is based solely on preceding information. Although SSM exhibits global modeling capabilities, it lacks sensitivity to long-range dependencies between non-adjacent pixels.

To address these challenges, we introduced the hybrid feature integration block (HFIB) and the unidirectional gating block (UGB). HFIB, positioned before the SS2D module, enhances both local and global feature extraction, enabling efficient integration with subsequent operations. The S6 module, as a core component of SS2D, performs feature extraction on the image blocks after scan expansion, followed by scan merging before being passed to the UGB module. UGB, which employs a gated CNN, captures

dependencies among adjacent features, dynamically selecting critical features while suppressing redundant information. Together, these enhancements improve the efficiency and accuracy of feature extraction.

Comparative experiments demonstrate substantial performance gains in small target detection. On the URPC2020 dataset's Test-A, our approach increased mAP50 and mAP50-95 by 7.1% and 4.8%, respectively, compared to the baseline model. On Test-B, the improvements were 7.2% and 5.1%. Furthermore, compared to the Mamba-YOLO model, our method achieved even greater enhancements on Test-B, with mAP50 and mAP50-95 increasing by 4.3% and 3.2%, respectively. These results underscore the Mamba model's potential for underwater detection tasks and highlight a promising direction for future research in this field. (Additional comparative data for models across various datasets are presented in Supplementary Tables 2 and 3).

### Underwater target detection network—UWNet

In underwater target detection, traditional methods often miss detections due to the complexity of underwater environments, which include numerous small targets, occlusions, and overlapping objects. These challenges have a substantial effect on accuracy and robustness. To overcome these challenges, we present a network architecture called UWNet, which builds on the high-performing YOLOv8 framework. Firstly, we replace the original downsampling convolution with SPDConv[31]. Traditional convolutions process the entire image directly, which can result in the loss of spatial details for small targets during downsampling. In contrast, SPDConv divides the input tensor into multiple subregions, enabling the network to extract features at a finer granularity, thereby improving small target detection. Additionally, by incorporating the MSDBlock into the backbone feature extraction section, UWNet achieves global feature extraction, overcoming the limitations of traditional CNNs that rely on local window-based modeling. The integration of the multi-scale implicit feature fusion (MSFF) with the detection head enables the network to consider information across different scales, leading to more comprehensive feature capture. The architecture of UWNet is illustrated in Fig. 3. Additionally, the CBS module processes the input through convolution, batch normalization, and the SiLU activation function.

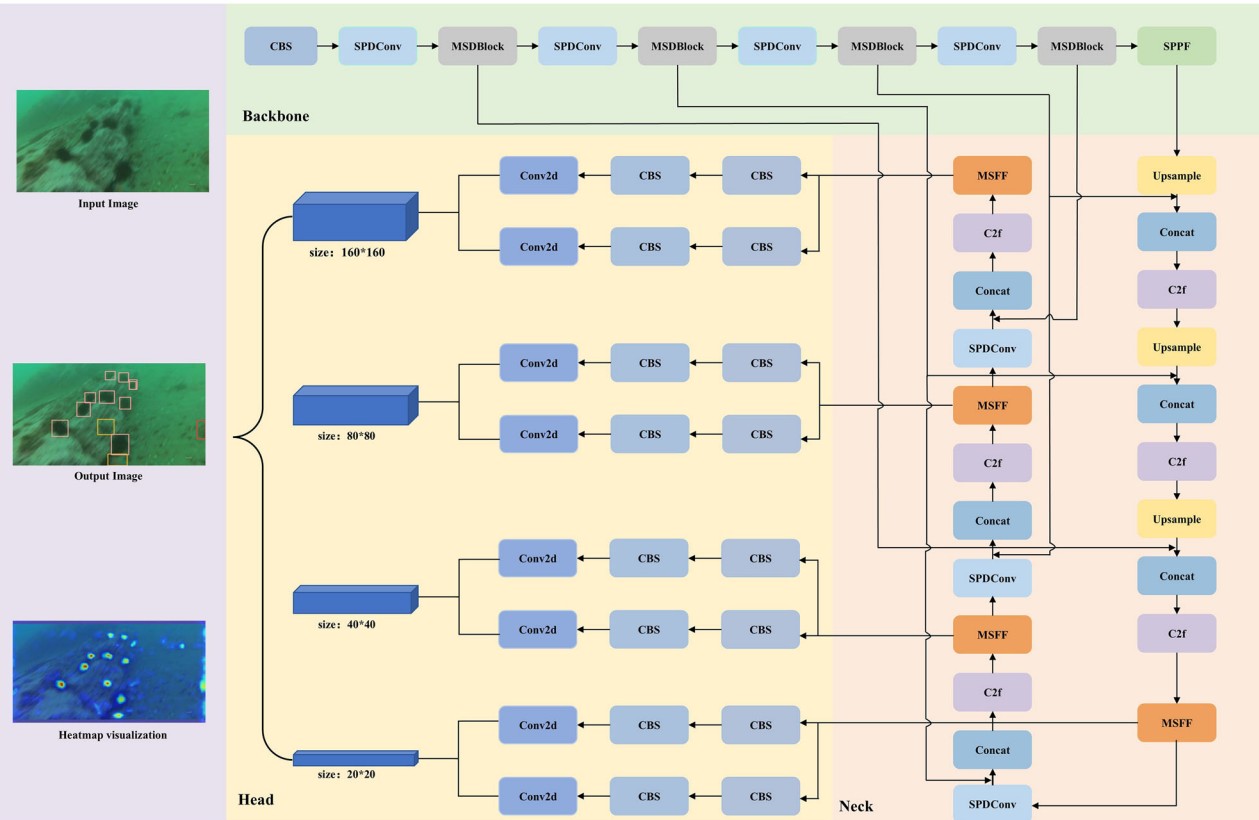

**Fig. 3 | The network architecture of the proposed UWNet model.**

We conducted experiments using the URPC2020 dataset from the National Underwater Robot Target Detection Algorithm Competition. This dataset comprises 5544 images, which were divided into training and validation sets using a 4:1 ratio. The dataset features four target categories: holothurian, echinus, scallop, and starfish. These targets are small and widely distributed in underwater images, making detection challenging. Our method achieved 86.5% mAP50 and 51.9% mAP50-95 on the validation set. Additionally, the precision and recall were 84.3% and 78.6%, respectively, showing improvements of 3.8%, 3.6%, 2.8%, and 2.2% compared to YOLOv8n. (The detailed data for Fig. 4 can be found in Supplementary Table 4). To further assess the effectiveness of our approach, we compared it with the latest object detection models. While accuracy in underwater object detection often depends on networks with greater width and depth, our model achieved state-of-the-art performance despite its remarkably low parameter count. This demonstrates that our method not only maintains high accuracy but also ensures model efficiency, making it suitable for deployment on resource-constrained underwater robots. The model has a total parameter count of only 6.67 million, and the final trained model size is 13.5 MB, showcasing its lightweight characteristics. Compared to YOLOv9, YOLOv10 and YOLOv11, UWNet's mAP50-95 is higher by 3.4%, 2.3%, and 1.1%, respectively. The Mamba-YOLO model, with a total parameter count of 21.8 million, achieved an mAP50-95 of 50.1% on the validation set, surpassing YOLOv10 and RT-DETR, and attaining a high mAP50 score. However, UWNet surpasses Mamba-YOLO while using less than one-third of its parameters, demonstrating the effectiveness of our proposed MSDBlock in underwater target detection scenarios. Overall, our model achieves superior performance in underwater target detection while remaining lightweight. The only area where our method is not optimal is GFLOPs, where it shows a slight gap compared to YOLOv5s and YOLOv9t. Nonetheless, the total parameter count of our method is lower than YOLOv5s, but the final trained UWNet model size is notably smaller than YOLOv9, as detailed in Supplementary Table 4. This is partly due to

Mamba's additional computational resources required for selective scanning. Despite this extra cost, it is justified by the substantial improvements in detection accuracy provided by the MSDBlock module, particularly in dynamic underwater environments and amidst color bias interference. In comparison to YOLOv5s and YOLOv9t, our method demonstrates a clear improvement in detection accuracy, especially in mAP50, where it exceeds YOLOv5 and YOLOv9 by 4% and 3.4%, respectively. Figure 4 illustrates the comparison between model parameter count and detection accuracy. Figure 4c illustrates the performance of different models across various underwater scenes. The blue boxes indicate correct detections, red boxes indicate incorrect detections, and purple boxes highlight missed targets. The first image shows a general underwater scene with sparsely distributed targets, making them easier to detect. The second image presents an underwater scene with occluded targets, where multiple targets are clustered together, increasing detection difficulty and the likelihood of false detections. The third image displays a scene with dense target distribution and occlusions, greatly raising detection difficulty. The visualization results clearly show that UWNet exhibits minor false detections at the edges of the first image, caused by the uniform color in underwater images, which makes it difficult to distinguish targets from the background at the edges. In the second image, UWNet detects 22 targets with very few errors compared to other models. In the third and fourth images, UWNet performs exceptionally well in high-difficulty detection scenarios, with only a few isolated false detections, which can be further minimized through image augmentation techniques.

**Performance on two test sets**

To validate the applicability of the proposed UWNet model in underwater small target scenarios, we evaluated its detection performance using two independent test sets in different environments. The selected datasets are the A and B test sets from the Underwater Target Detection Algorithm Competition URPC2020, containing 800 and 1200 test

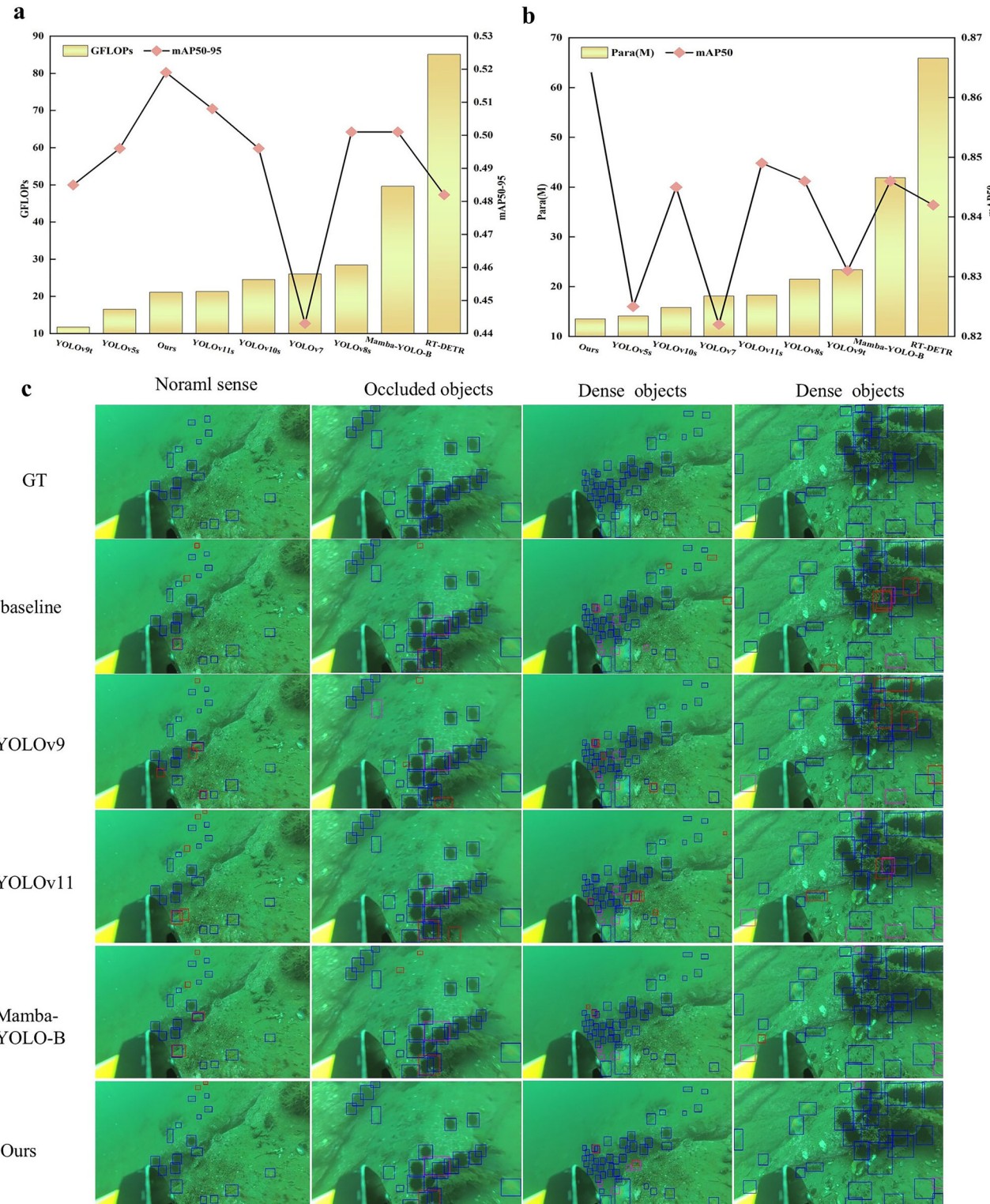

**Fig. 4 | Performance of our deep learning approach versus other models.**
**a** Comparison of mAP50-95 and GFLOPs values on the validation set across different models. **b** Comparison of mAP50 and parameter count (Para) values on the validation set across different models. **c** Detection results of different models in various underwater scenarios.

images, respectively. During the experiments, we used the best weights obtained by training UWNet on the URPC2020 dataset to evaluate these two datasets. Experimental results indicate that on the A test set, our method improved mAP50 and mAP50-95 by 7.1% and 4.8%, respectively, compared to the baseline model. Results on the B test set also revealed notably improvements in detection accuracy, with mAP50 and

mAP50-95 increasing by 7.2% and 5.1%, respectively, compared to the baseline model. We conducted experiments using the latest object detectors based on the YOLO series, Mamba, and Transformer architectures on both datasets, as shown in Fig. 5. The results demonstrate that our model achieved the highest mAP50 and mAP50-95 values on Test A and Test B.

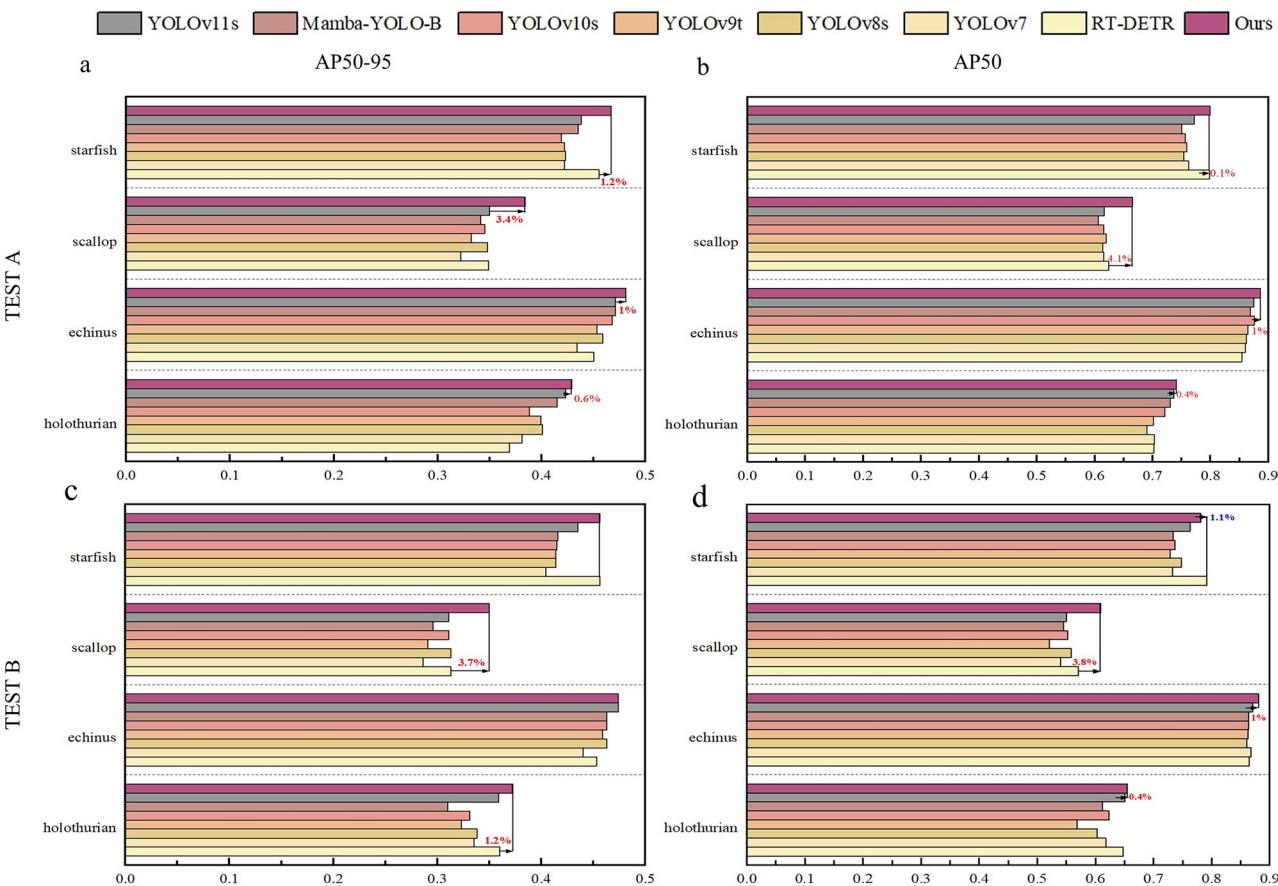

**Fig. 5 | Performance of our deep learning approach versus other models.** Two different test sets were selected to validate the performance and generalization ability of various models in different scenarios. **a** AP50-95 values of the models on the Test A dataset. **b** AP50 values of the models on the Test A dataset. **c** AP50-95 values of the models on the Test B dataset. **d** AP50 values of the models on the Test B dataset.

In underwater detection scenarios, echinus is entirely black and resembles rocks in underwater images, making it easy to confuse with the background. However, due to their abundance, they are relatively easier to detect. Scallops, on the other hand, often occupy a very small area in underwater images, and their shape resembles the seabed, leading to frequent false positives and missed detections by many models. Although starfish have a small resolution, they are not as densely distributed as scallops and have bright surface colors, making them relatively easier to detect compared to scallops. Holothurians, while having a larger resolution compared to scallops and starfish, exhibit non-uniform shapes in images, as they tend to change form when startled, making them more challenging to detect consistently.

On Test A and Test B, UWNet attained the highest detection accuracy for holothurians, an irregularly shaped target class, with mAP50 scores of 0.741 and 0.655, respectively. For echinus, the most sample-diverse class, UWNet effectively captured key features, achieving mAP50 detection accuracies of 88.6% and 88.1% on the two test sets. Additionally, UWNet excelled in detecting densely distributed scallops, outperforming all comparison models in detection accuracy. While UWNet's mAP50 performance on starfish was marginally lower than RT-DETR, it achieved the highest precision under the stricter mAP50-95 evaluation metric.

In comparison, the RT-DETR model, with ResNet50 as its backbone, has 32.66 million parameters—390% more than UWNet. Furthermore, RT-DETR introduces substantial computational complexity, making it unsuitable for real-time underwater robotics. More critically, RT-DETR demonstrated poor generalization across underwater datasets, highlighting its lack of robustness in underwater detection tasks. Detailed

performance data for the models on Test A and Test B are available in Supplementary Table 5.

## Validating the generalization of UWNet across different underwater scenarios

To assess UWNet's generalization across various underwater environments and water quality conditions, experiments were conducted on two additional datasets: DUO and URPC2021. The DUO dataset includes 6671 training images and 1111 test images, while the URPC2021 dataset comprises 7600 images split into training and validation sets in a 4:1 ratio. UWNet was trained separately on each dataset, with hyperparameters iteratively optimized based on experimental outcomes to enhance performance.

On the DUO dataset, UWNet achieved mAP50 and mAP50-95 scores of 87.1% and 69.5%, surpassing state-of-the-art object detectors. Compared to the baseline model, UWNet increased mAP50 and mAP50-95 by 4% and 6%, respectively. Notably, UWNet recorded the highest AP50 values across all target categories: 89.3% for holothurian, 93.5% for echinus, 71% for scallop, and 94.4% for starfish.

On the URPC2021 dataset, UWNet achieved mAP50 and mAP50-95 scores of 85.5% and 52.4%, achieving the highest accuracy among all comparative models. UWNet exhibited superior detection for holothurian, scallop, and starfish, with its echinus detection trailing YOLOv11 by just 0.2%. Notably, YOLOv11 has a deeper network and larger parameter count, with 9.41 million parameters compared to UWNet's 6.67 million. Furthermore, after training, YOLOv11's model size is 18.5 MB, compared to UWNet's 13.5 MB. This highlights UWNet's ability to maintain high

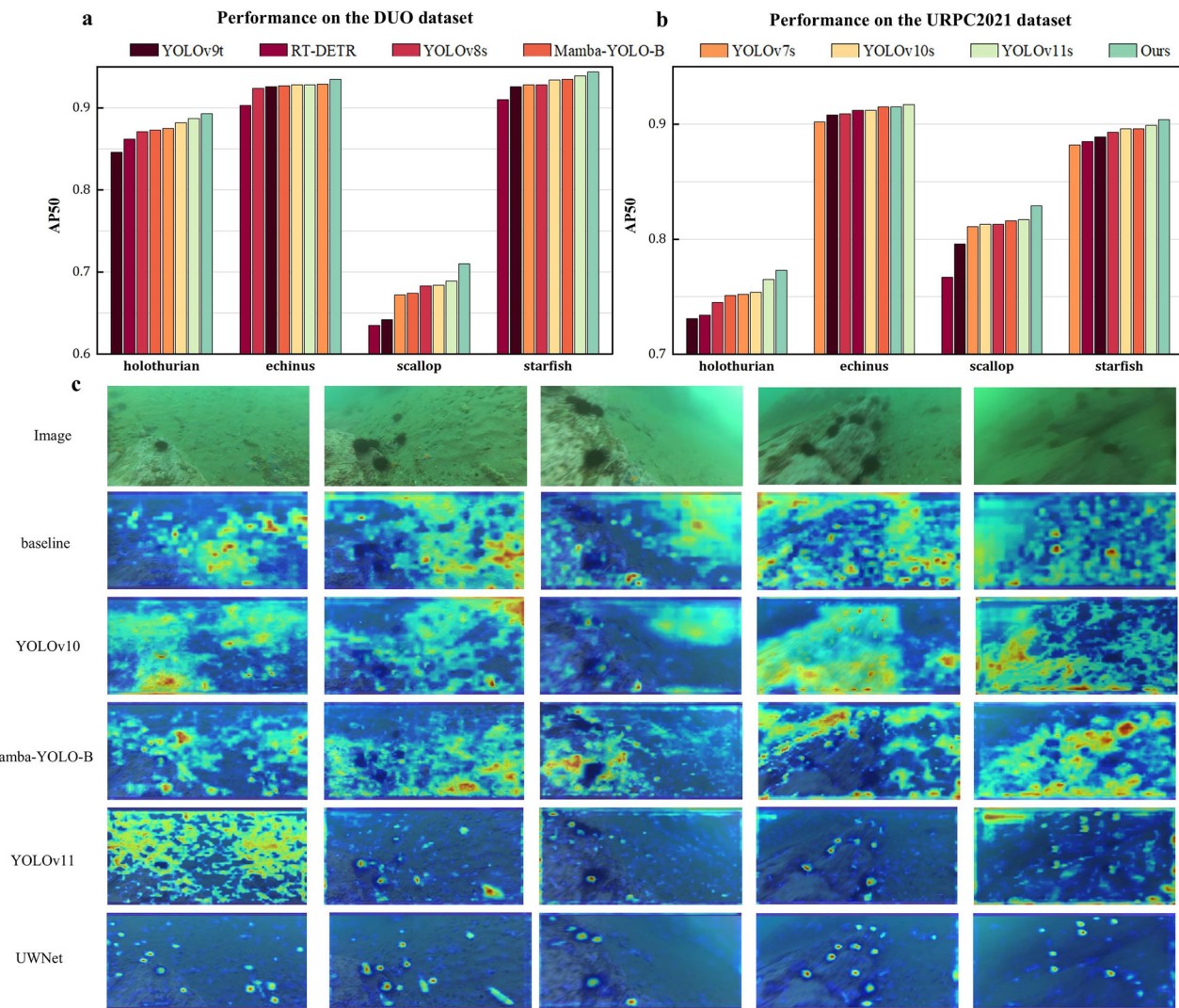

**Fig. 6 | Heatmaps of feature extraction in various models. a** Performance of different models on the DUO datasets. **b** Performance of different models on the URPC2021 datasets. **c** Visualization of heatmap effects during feature extraction using the GradCAM method. The left side shows the original input image, followed by the visualization results of the baseline, YOLOv10, Mamba-YOLO, and UWNet models.

detection accuracy while reducing model complexity and storage needs, making it ideal for underwater applications.

The experiments on the DUO and URPC2021 datasets validate UWNet's effectiveness in underwater target detection and robust generalization across datasets. To further evaluate the model's performance, GradCAM was employed to generate heatmaps, visually comparing UWNet's focus on objects and regions during feature extraction with other models. Visualizations of these feature maps are shown in Fig. 6.

## Discussion

Underwater target detection is currently a major area of interest among researchers, who are addressing two main challenges in this field. First, existing underwater datasets often suffer from image blurring and color distortion[32–34]. Underwater images frequently exhibit a blue or green color cast due to light absorption and scattering, making it difficult to differentiate between targets and the background. Second, underwater targets are often small and can overlap, complicating the detection of all targets simultaneously. To address these challenges, it is crucial to develop a precise and lightweight underwater target detection network. Such a network should achieve accurate detection results with minimal parameter cost, making it suitable for deployment on underwater robots. This advancement would greatly enhance detection accuracy while satisfying real-time requirements,

thereby enhancing marine exploration. Detecting and tracking underwater organisms in real time, along with monitoring population trends, are crucial for evaluating the stability of marine ecosystems and aiding scientific research.

In Fig. 7, we analyzed the size distribution of targets across various underwater datasets. The analysis reveals a common trend: echinus are the most frequently annotated small-sized targets, while holothurians predominantly appear as medium or large-sized objects in underwater scenes. For detailed data on each dataset, refer to Supplementary Table 6. UWNet exhibits strong performance in detecting echinus, achieving AP50 scores of 88.6%, 88.1%, 91.5%, and 93.5% on the URPC2020 test sets, the URPC2021 dataset, and the DUO dataset, respectively. Figure 7 also shows that scallops and starfish are the next most abundant small targets after echinus in the URPC2020 and URPC2021 datasets. However, the detection performance for scallops is relatively lower compared to other targets. This reduced performance may be due to the small size of scallops in images, making them difficult to distinguish from the seabed in the blue and green-dominated underwater backgrounds, which hampers the model's feature extraction capabilities. Despite this, UWNet considerably outperforms other models in detecting scallops and starfish, further demonstrating its superior capability in handling small underwater targets. Overall, UWNet achieves optimal results across all three underwater datasets, underscoring its robustness in

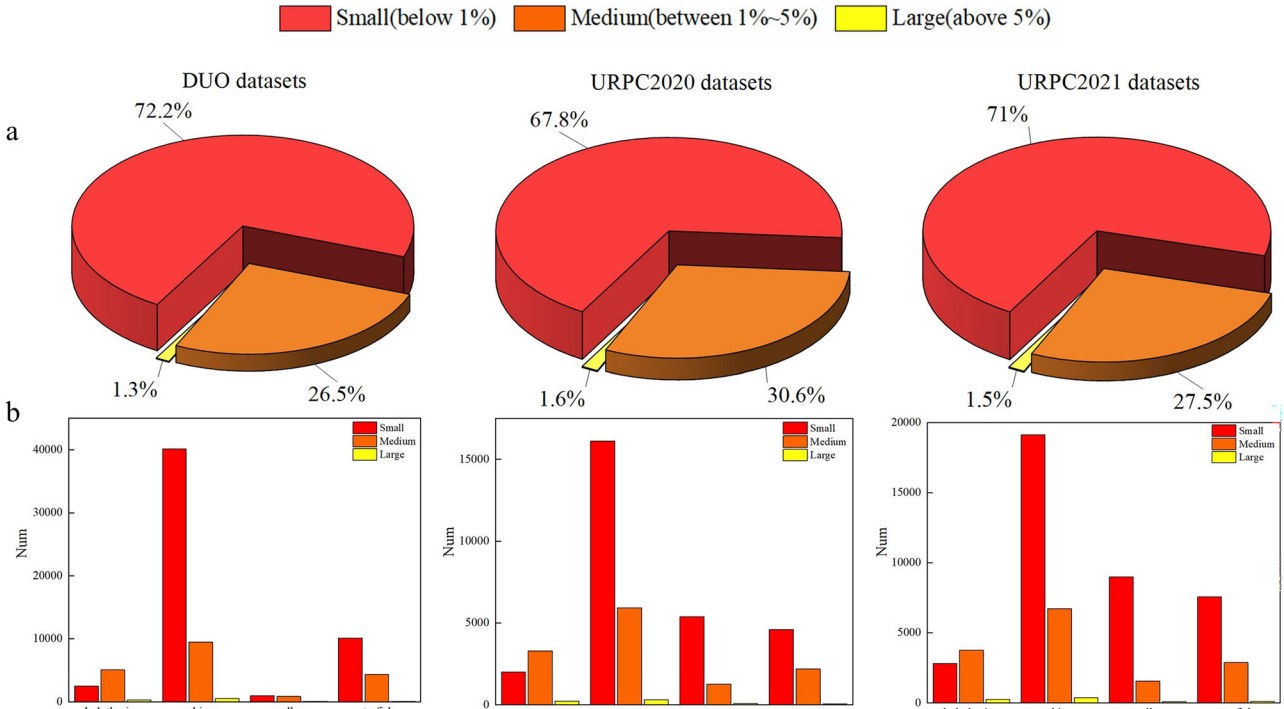

**Fig. 7 | Statistics of dataset label box sizes. a** The distribution of annotated bounding box sizes across different datasets, from left to right: DUO, URPC2020, and URPC2021 datasets. **b** The distribution of bounding box sizes for different categories in the DUO, URPC2020, and URPC2021 datasets. The categories are presented in the following order (from left to right): holothurian, echinus, scallop, and starfish.

various underwater scenarios. The experimental findings effectively showcase the Mamba model's superior capability in handling small underwater targets. Our method has the following advantages: Firstly, we have designed a feature extraction module—MSDBlock, which effectively integrates the Mamba model into the underwater detection scenario, addressing the issue of information loss for small targets during local window feature extraction in CNN methods. The Mamba model establishes long-distance dependencies through HiPPO and selectively retains or discards input information by parameterizing the model's input. Secondly, we have introduced a module—the multi-scale implicit feature fusion module (MSFF). This module, through multi-perspective feature extraction, comprehensively extracts feature information of the input image at different receptive fields. Subsequently, it implicitly increases the feature dimensions, mapping the input features to higher dimensions, thereby enhancing the model's learnability. Through the MSFF module, the model can better differentiate between target and background information.

Lastly, addressing the challenges faced in current underwater target detection, we have proposed a network framework—UWNet. During the experiment, we focused on the network architecture design while iteratively optimizing the hyperparameters across three datasets based on the experimental results, ensuring that the model achieved optimal performance.

Comparing with other state-of-the-art models, YOLOv9 introduced programmable gradient information (PGI) and auxiliary reversible branches, which considerably enhanced detection accuracy. However, this enhancement was achieved at the expense of increased model parameters and longer training durations. YOLOv10 addressed the latency issue caused by Non-Maximum Suppression (NMS) and enabled end-to-end object detection. While both models perform well in general scenarios, they still require further refinement for underwater target detection applications. YOLOv11, the latest object detector released in 2024, features a more efficient feature extraction module called C3k2, designed to enhance the model's feature extraction capabilities. Additionally, a self-attention mechanism was incorporated in the last layer of the backbone to improve the model's global perception ability. RT-DETR combines the strengths of

CNNs and Transformers, utilizing the global modeling capabilities of Transformers for object detection tasks. However, a major limitation of RT-DETR is its large parameter size and slow inference speed, making it less suitable for real-time detection. Additionally, its performance in detecting small underwater targets is suboptimal, primarily because Transformers rely on large datasets for training, and specialized underwater small target data is often scarce. In contrast, this paper introduces UWNet, which combines the strengths of Mamba and CNNs. UWNet attains optimal detection accuracy while maintaining the lowest number of parameters, outperforming current state-of-the-art detectors in underwater target detection.

Our future research will focus on two key aspects. First, although the current model's parameter size is relatively efficient, further optimization can be achieved through model pruning and knowledge distillation to create an even more lightweight underwater target detector. Second, we aim to explore two underwater data augmentation methods: one involves using underwater image processing techniques to enhance image clarity and reduce the impact of color distortion; the other leverages generative AI models, such as the Diffusion model, to augment underwater datasets. This would allow the model to learn from various colors and backgrounds by expanding the available data for underwater target detection. Additionally, we plan to evaluate UWNet's applicability for detecting small targets in non-underwater environments, further validating its versatility across different application scenarios. Through systematic experiments and performance optimization, we anticipate that UWNet will demonstrate robustness and flexibility in a wide range of object detection tasks.

The stability of marine ecosystems is closely tied to the sustainable development of countries worldwide. However, human exploitation of natural resources has gradually disrupted this balance. To address this issue, quantitative monitoring of marine organisms is essential for assessing the stability of marine ecosystems in real-time. The technology for underwater target detection is crucial in this process, as it effectively identifies and records both the species and the abundance of marine organisms. providing reliable data to support marine conservation efforts. Our proposed method, UWNet, is a lightweight and high-precision network architecture. When

integrated with underwater robots, it enables efficient detection and precise quantification of underwater organisms, offering scientists immediate data to assess changes in the marine environment. Additionally, UWNet can be used to track rare underwater species, facilitating their protection in real time. By continuously monitoring the movements and behaviors of these species, scientists can better understand their survival status and implement appropriate conservation measures to prevent population decline. In conclusion, underwater target detection technology is critical for maintaining the stability of marine ecosystems. Its application will inject new energy into global marine conservation efforts, fostering a harmonious coexistence between humanity and nature.

## Methods
### State space models
Due to its strong ability to capture long-range dependencies and efficiently represent dynamic systems, the structured state space model (SSM) has garnered increasing attention from researchers. SSM is conceptually similar to recurrent neural networks (RNNs), with the primary distinction being that SSM removes the nonlinear transformation component from the hidden state update equations. At its core, SSM is represented by a set of linear ordinary differential equations, as shown in Eq. (1):

$$h'(t) = Ah(t) + Bx(t)$$
$$y(t) = Ch(t) \tag{1}$$

Where $A$ is the state transition matrix, representing the relationship of the hidden state $h(t)$ as it evolves over time. The input matrix B and output matrix $C$ represent the relationships between the input signal $x(t)$, hidden state $h(t)$, and output $y(t)$, respectively. However, in deep learning applications, the given signals are often discrete. This necessitates converting the state-space equations from a continuous system to a discrete system. This conversion is one of the key improvements in the evolution of SSM to S4—parameter discretization. Specifically, this is achieved by applying a zero-order hold to the input signal. The rules for parameter discretization are shown in Eq. (2):

$$\bar{A} = \exp(\Delta A)$$
$$\bar{B} = (\Delta A)^{-1}(\exp(\Delta A) - I)\Delta B \tag{2}$$

After applying the discretization rules, the discrete SSM representation is given as shown in Eq. (3):

$$h'_t = \bar{A}h_{t-1} + \bar{B}x_t$$
$$y_t = Ch_t \tag{3}$$

The final Mamba model incorporates a selective scanning mechanism into the SSM, allowing the parameters A–C in the state-space equation to become input-dependent parameters (which can influence the state transition matrix A through the discretization rules). This mechanism is particularly important for underwater small target detection. Its core advantage lies in scanning information from different directions, enabling a comprehensive understanding of the input data within both the current and global context. This allows the Mamba model to dynamically adjust the parameter matrices.

In underwater target detection, UWNet incorporates the Mamba model to address the unique challenges posed by underwater environments. Underwater images often suffer from substantial color distortion, predominantly in blue and green tones, which complicates the distinction between targets and the background. Additionally, the presence of numerous small and frequently overlapping targets poses formidable challenges to traditional CNN-based local feature extraction methods. To address these issues, we propose the MSDBlock module within the UWNet model. This module seamlessly integrates CNN and SSM techniques, enabling precise local feature extraction while effectively capturing global features.

The MSDBlock incorporates a lightweight attention mechanism called the Hybrid Feature Integration Block (HFIB), which optimizes attention on target regions during local feature extraction. The local features extracted by HFIB are subsequently embedded into the SSM module, which leverages selective scanning to model long-term dependencies in underwater images, capturing deep global scene information. By progressively modeling features from local to global scales, the MSDBlock provides subsequent network layers with rich and structured feature representations. In UWNet, the MSDBlock serves as the backbone feature extraction module, enhancing the network's ability to represent features for complex underwater targets.

### Implicit feature mapping
Element-wise multiplication is an effective mechanism for feature fusion. By directly multiplying features element by element from different subspaces, it achieves implicit high-dimensional nonlinear feature mapping without increasing the network's depth or width, thus controlling model complexity. Moreover, this operation exhibits behavior analogous to polynomial kernels in kernel methods, enhancing the feature space's representational power while ensuring computational efficiency. As a result, it offers a practical solution for deep learning models dealing with high-dimensional feature representations.

To address the challenge of distinguishing objects from complex underwater backgrounds, this paper introduces an implicit multi-scale high-dimensional mapping method called the MSFF module. In the multi-scale feature extraction stage, the MSFF module first employs average pooling on the input feature maps, reducing redundancy while preserving critical information. Subsequently, a set of parallel depthwise separable convolutions extracts multi-scale features, where each branch performs convolutions along horizontal and vertical directions separately, effectively capturing fine-grained features at multiple scales.

During the feature fusion stage, the MSFF module uses element-wise multiplication to fuse multi-scale features implicitly, projecting them into higher-dimensional spaces to improve feature distinguishability for object-background separation. To enhance the fused features further, a Sigmoid activation function is applied after the element-wise multiplication to improve the non-linear expressiveness of the features. Finally, a residual connection integrates the optimized fused features with the original input features, enabling efficient cross-layer information exchange. I n UWNet, the MSFF module is strategically positioned before each detection head, enhancing the model's capacity to accurately identify and capture underwater targets.

### Model architecture and training
Our deep learning model is built upon the YOLO framework, comprising three core elements: the Backbone, the Neck, and the Head. The SPDConv and MSDBlock modules are integrated into the Backbone to enhance the model's feature extraction capabilities from input images. The MSFF module is added before each detection head in the Head section to improve the model's ability to detect targets of different sizes. For the URPC2020 dataset, the batch size was set to 8, while for the URPC2021 and DUO datasets, the batch size was set to 16. The input image size during training was $640 \times 640$, and the number of training epochs was set to 200. We employed the built-in image augmentation strategies provided by YOLOv8, including mosaic, fliplr, and translate. The model was trained using the SGD optimizer, with both the initial learning rate (lr0) and final learning rate (lrf) set to 0.01, and the initial momentum set to 0.937. In this study, the experiments were conducted using PyTorch 2.0.0 and Cuda 11.8. All training, validation, and testing on the URPC2020 dataset was done on an NVIDIA RTX 2080Ti. Due to the large number of training images in the URPC2021 and DUO datasets, the training, validation, and testing processes on both datasets were performed on two NVIDIA RTX 2080Ti. Specifically, we trained the model using the URPC2020, URPC2021, and DUO datasets, and tested the model's performance on each dataset. The weights obtained from training on URPC2020 were used to evaluate the model on the Test A

and Test B datasets. (The detailed model training hyperparameters are provided in Supplementary Table 10).

## Validation metrics on underwater datasets

We assessed the performance of various models on underwater datasets by utilizing the following metrics: Accuracy, Recall, Average Precision (AP), mean Average Precision (mAP), Floating Point Operations (GFLOPs), and parameter count (Para, M). The calculation methods for Accuracy (P), Recall (R), AP, and mAP are provided in Eq. (4):

$$\text{Precision} = \frac{TP}{TP + FP}$$
$$\text{Recall} = \frac{TP}{TN + FN}$$
$$AP = \int_0^1 P(R)dR \qquad (4)$$
$$mAP = \frac{1}{n}\sum_{i=1}^{n} AP_i$$

In the formulas mentioned above, TP stands for true positives, TN for true negatives, FP indicates false positives, and FN signifies false negatives. AP indicates the detection precision for a specific target class, while mAP represents the average detection precision across all classes. These two metrics are commonly used to assess the accuracy of detection models. GFLOPs refers to the number of floating-point operations required to process a single input image; a lower value indicates a less complex model. GFLOPs and Para (parameter count) are typically used to evaluate a model's lightweight performance, with lower values indicating a more lightweight and efficient model.

## Data availability

The underwater datasets used in this study (URPC2020, URPC2021, and DUO) have all been uploaded to the GitHub repository.

## Code availability

The code for this study can be found at https://github.com/JML123123/UWNet.

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

## Acknowledgements

This research was supported in part by the National Natural Science Foundation of China (62403108, 42301256, U20A20197, and 61973063), the Liaoning Provincial Natural Science Foundation Joint Fund (2023-MSBA-075), the Ministry of Industry and Information Technology Project (TC220H05X-04), the Scientific Research Foundation of Liaoning Provincial Education Department (LJKQR20222509), the Fundamental Research Funds for the Central Universities (N2426005).

## Author contributions

Ideas were developed by Y.M.Z, L.L, C.D.W, W.C, and Z.L.L. Methodology was designed by Y.M.Z, J.M.L, L.L, and C.D.W. The underwater dataset was collected by J.M.L, H.Y.Z, Z.R.F, and L.Y.M. The model code was written and debugged by Y.M.Z, J.M.L, H.Y.Z, and L.Y.M. Model training and data organization were conducted by H.Y.Z and L.Y.M. Model results were visualized by Y.M.Z, J.M.L, L.Y.M, Z.R.F, and L.L. Model comparison experiments and ablation studies were performed by Y.M.Z, J.M.L, and H.Y.Z. Manuscript formatting was managed by H.Y.Z, L.Y.M, W.C, and Z.L.L. All authors made substantial contributions to the paper content and have read and approved its publication.

## Competing interests

The authors declare no competing interests.
