## [Peer Review File · Communications Engineering]

This manuscript has been previously reviewed at another journal. This document only contains information relating to versions considered at Communications Engineering.

A deep learning framework based on Structured Space Model for detecting small objects in complex underwater environments

Corresponding Author: Professor Yaoming Zhuang

Version 0:

Reviewer comments:

Reviewer #1

(Remarks to the Author)

The author has meticulously revised the manuscript in accordance with the reviewer's comments, and it has generally met the standards for publication. The manuscript repeatedly mentions that the proposed method is capable of realizing real-time detection on the platform of underwater robots. However, Table 1 indicates that the GFLOPs of the proposed method is approximately 21.1, suggesting that it may be challenging to achieve real-time detection on underwater robots. It is recommended to avoid the claim of real-time detection if possible. Furthermore, it is suggested that the English expressions within the manuscript be further refined.

Northeastern University,
Shenyang, 110819, China

Response from Authors to Reviewers

Submission Number: COMMSENG-24-0641-T

Article Title: A deep learning framework based on SSM for detecting small and occluded objects in complex underwater environments

Dear Editor and Reviewers,

We greatly appreciate your recognition of our work and your valuable comments. In response to the reviewers' feedback, we will make further detailed revisions to our manuscript titled "A deep learning framework based on SSM for detecting small and occluded objects in complex underwater environments" (Submission Number: COMMSENG-24-0641-T) and carefully review our supplementary files to ensure they meet the journal's requirements. In our response letter, we have highlighted the key areas for your attention in yellow, corresponding to sections in the manuscript with changes. We sincerely thank you for the effort you have dedicated to our work and hope that these revisions will meet with your approval.

Reviewer 1:

1. The author has meticulously revised the manuscript in accordance with the reviewer's comments, and it has generally met the standards for publication. The manuscript repeatedly mentions that the proposed method is capable of realizing real-time detection on the platform of underwater robots. However, Table 1 indicates that the GFLOPs of the proposed method is approximately 21.1, suggesting that it may be challenging to achieve real-time detection on underwater robots. It is recommended to avoid the claim of real-time detection if possible. Furthermore, it is suggested that the English expressions within the manuscript be further refined.

Authors' response:

We sincerely thank you for your valuable comments and suggestions on our manuscript. Your insights have been instrumental in enhancing both the clarity and accuracy of our work. We acknowledge that our previous claim regarding real-time detection may have caused some confusion. After careful consideration, we have revised the manuscript to more accurately describe our method as a lightweight detection approach for underwater robots. In addition, we have taken your suggestion to refine the English expressions within the manuscript. We have thoroughly reviewed and edited the text.

The modified content in the manuscript is highlighted in red. The modifications to the Abstract section correspond to page 1 of the manuscript; the modifications to the Introduction section correspond to page 2 of the manuscript; the modifications to the Results section correspond to pages 5 and 9; and the modifications to the Methods section correspond to pages 12.

We use blue font to indicate revisions to the English expressions in the manuscript. The modifications to the Introduction section correspond to page 1 of the manuscript; the modifications to the Result section correspond to page 2, 3, 4, 5, 7 and 8; and the modifications to the Methods section correspond to pages 13.

Thank you again for your feedback.